# Riemannian Stochastic Gradient Descent for Tensor-Train Recurrent Neural Networks

## Abstract

The Tensor-Train factorization (TTF) is an efficient way to compress large weight matrices of fully-connected layers and recurrent layers in recurrent neural networks (RNNs). However, high Tensor-Train ranks for all the core tensors of parameters need to be element-wise fixed, which results in an unnecessary redundancy of model parameters. This work applies Riemannian stochastic gradient descent (RSGD) to train core tensors of parameters in the Riemannian Manifold before finding vectors of lower Tensor-Train ranks for parameters. The paper first presents the RSGD algorithm with a convergence analysis and then tests it on more advanced Tensor-Train RNNs such as bi-directional GRU/LSTM and Encoder-Decoder RNNs with a Tensor-Train attention model. The experiments on digit recognition and machine translation tasks suggest the effectiveness of the RSGD algorithm for Tensor-Train RNNs.

## 1 Introduction

Recurrent Neural Networks (RNNs) are typically composed of large weight matrices of fully-connected and recurrent layers, thus massive training data as well as exhaustive computational resources are required. The Tensor-Train factorization (TTF) aims to reduce the redundancy of RNN parameters by reshaping large weight matrices into high-dimensional tensors before factorizing them in a Tensor-Train format Oseledets (2011). The notation of Tensor-Train usually suggests that TTF is applied for the tensor representation of model parameters. Tensor-Train was initially applied to fully-connected layers Novikov et al. (2015), and it has been recently generalized to recurrent layers in RNNs such as LSTM and GRU Yu et al. (2017). Compared with other tensor decomposition techniques like the CANDECOMP/PARAFAC decomposition Kolda & Bader (2009) and Tucker decomposition Kim & Choi (2007), Tensor-Train can be easily scaled to arbitrarily high dimensions and have the advantage of computational tractability to significantly large weight matrices.

Given a vector of Tensor-Train ranks $\boldsymbol{r} = (r_1, r_2, \cdots, r_{d+1})$, TTF decomposes a $d$-dimensional tensor $\mathbf{W} \in R^{(m_1 \cdot n_1) \times (m_2 \cdot n_2) \times \cdots \times (m_d \cdot n_d)}$ into a multiplication of core tensors according to (1), where the $k$-th core tensor $\mathbf{C}^{[k]} \in R^{r_k \times m_k \times n_k \times r_{k+1}}$, and any index pair $(i_k, j_k)$ satisfies $1 \leq i_k \leq m_k, 1 \leq j_k \leq n_k$. Additionally, the ranks $r_1$ and $r_{d+1}$ are fixed to 1.

$$\mathbf{W}((i_1, j_1), (i_2, j_2), ..., (i_d, j_d)) = \mathbf{C}^{[1]}(r_1, i_1, j_1, r_2)\mathbf{C}^{[2]}(r_2, i_2, j_2, r_3)\cdots\mathbf{C}^{[d]}(r_d, i_d, j_d, r_{d+1}) \quad (1)$$

Thus, when the TTF technique is applied to the fully-connected (FC) layer with feed-forward weight matrix $\boldsymbol{W}$, a tensor $\mathbf{W}$ is firstly converted from $\boldsymbol{W}$ and is then decomposed to a multiplication of the core tensors as shown in (2), where $\mathbf{C}^{[t]}$ is the $t$-th core tensor, $\mathbf{X}$ denotes a tensor of input, $\mathbf{B}$ refers to a tensor of bias, the tensor of outputs $\hat{\mathbf{Y}} \in R^{n_1 \times n_2 \times \cdots \times n_d}$, and $\sigma$ is a sigmoid function. For clarity, the notation $TTL(\boldsymbol{W}, \boldsymbol{X})$ is used to simplify the representation of a Tensor-Train fully-connected layer, which is shown in (3).

$$\hat{\mathbf{Y}}(j_1, j_2, ..., j_d) = \sigma(\sum_{i_1=1}^{m_1} \cdots \sum_{i_d=1}^{m_d} \prod_{t=1}^{d} \mathbf{C}^{[t]}(r_t, i_t, j_t, r_{t+1})\mathbf{X}(i_1, i_2, ..., i_d) + \mathbf{B}(j_1, j_2, ..., j_d)) \quad (2)$$

$$\hat{\mathbf{Y}} = \sigma(TTL(\boldsymbol{W}, \boldsymbol{X})). \quad (3)$$

Likely, we use (4) to represent an RNN with a feed-forward weight matrix $\boldsymbol{W}$ and a recurrent weight matrix $\boldsymbol{U}$. In (4), $\boldsymbol{X}_t$ is an input matrix at time $t$, $\boldsymbol{h}_{t-1}$ and $\boldsymbol{h}_t$ separately denote the hidden vectors of time $t-1$ and $t$, and $\sigma$ refers to the sigmoid function.

$$\boldsymbol{h}_t = \sigma(TTL(\boldsymbol{W}, \boldsymbol{X}_t) + TTL(\boldsymbol{U}, \boldsymbol{h}_{t-1})). \tag{4}$$

The largest benefit from Tensor-Train models is the capability to reduce the model parameters tremendously. For example, a Tensor-Train FC layer needs $\sum_i m_i \cdot n_i \cdot r_i \cdot r_{i+1}$ parameters in total. In comparison, the total number of parameters of an FC layer is about $O(\prod_i m_i \cdot \prod_i n_i)$, which is much larger than the associated Tensor-Train FC one.

The Tensor-Train models are found widespread. For example, Yang et al. (2017) set up Tensor-Train Recurrent Neural Networks for video classification, and Yu et al. (2017) applied the Tensor-Train as an End-to-End dynamic model for multi-variate forecasting environmental data. However, those applications focused on simple deep learning architectures, and a vector of Tensor-Train ranks $\mathbf{r}$ is element-wise fixed. When more complex deep models with numerous Tensor-Train layers are involved, it is not easy to find appropriate Tensor-Train ranks for each core tensor. However, when applying a vector of shared and high Tensor-Train ranks to all parameters of Tensor-Train layers, each of them has a vector of lower Tensor-Train ranks, which results in an additional decrease in the number of parameters.

This work applies Riemannian Stochastic Gradient Descent (RSGD) to the RNN Tensor-Train layers. Unlike Stochastic Gradient Descent (SGD), which conducts the update of parameters in the Euclidean space, RSGD finds optimal core tensors of parameters associated with a vector of lower Tensor-Train ranks in Riemannian Manifold, thereby decreasing the number of total parameters of Tensor-Train RNNs.

Our work is inspired by Bonnabel (2013) and Lubich et al. (2015), which are two excellent introductions to the theory of Stochastic Gradient Descent on Riemannian Manifolds. Additionally, Zhang et al. (2016) was a recent work that proposed a fast stochastic optimization on Riemannian Manifolds. Besides, Absil et al. (2009) is a useful reference of Riemannian optimization.

**Contributions**. We summarize the key contributions of this paper as follows:

- This work applies RSGD to iteratively find vectors of lower Tensor-Train ranks with the update of parameters in the training process. The RSGD algorithm and the related theoretical analysis are also presented.
- We design Bi-directional Tensor-Train GRU/LSTM Chung et al. (2014), and Encoder-Decoder Tensor-Train RNNs with a Tensor-Train Attention mechanism Luong et al. (2015). We apply the RSGD algorithm to the Tensor-Train RNN models on the digit recognition and machine translation tasks.

To the best of our knowledge, this is the first work that applies RSGD to train Tensor-Train RNNs to find the optimal core tensors of parameters with vectors of lower Tensor-Train ranks. Moreover, this is the first work that builds Tensor-Train RNNs with complex architectures for natural language processing (NLP) tasks.

## 2  RIEMANNIAN STOCHASTIC GRADIENT DESCENT

The optimization problem of Tensor-Train RNNs can be formulated as a Riemannian optimization problem as shown in (5), where $\{\boldsymbol{X}, \boldsymbol{Y}\}$ is a data sequence with length $T$, $\mathbf{W}$ represents a tensor of parameters which lies in a $d$-dimensional Riemannian Manifold $(M, \mu)$ with a Riemannian measure $\mu$, and the Tensor-Train ranks for core tensors of $\mathbf{W}$ must be element-wise no higher than the vector $\mathbf{r} = (r_1, r_2, ..., r_{d+1})$ as shown in (5).

$$\min_{\mathbf{W} \in M} f(\mathbf{W}; \boldsymbol{X}, \boldsymbol{Y})$$
$$s.t., tt\_rank(\mathbf{W}) \leq \mathbf{r}. \tag{5}$$

Besides, the Riemannian measure $\mu$ induces an inner product structure in each tangent space $T_{\boldsymbol{x}}M$ associated with a tensor $\boldsymbol{x} \in M$. Specifically, $\forall \mathbf{u}, \mathbf{v} \in T_{\boldsymbol{x}}M$, the inner product $< \mathbf{u}, \mathbf{v} > = \mu_{\boldsymbol{x}}(\mathbf{u}, \mathbf{v})$.

---

**Algorithm 1** Riemannian Stochastic Gradient Descent

---
**1.** Given the labeled input data $(\boldsymbol{X}, \boldsymbol{Y})$ with sequence length $T$, and the learning rate $\eta$.
**2.** $\mathbf{W}^{(0)} \leftarrow$ the randomly initialized core tensors $\{\mathbf{C}^{[1]}, \mathbf{C}^{[2]}, ..., \mathbf{C}^{[d]}\}$.
**3.** For $t = 1, 2, ..., T$:
**4.**      For $i = 1, 2, ..., d$:
**5.**           Choose a gradient $\mathbf{g}_{\mathbf{C}^{[i]}} = \nabla_{\mathbf{C}^{[i]}} f(\mathbf{W}^{(t)}; \boldsymbol{X}, \boldsymbol{Y})$ in the tangent space $T_{\mathbf{C}^{[i]}} M$.
**6.**           $\hat{\mathbf{A}}_i = \mathbf{C}^{[i]} - \eta \mathbf{g}_{\mathbf{C}^{[i]}}$.
**7.**           $\hat{\mathbf{C}}^{[i]} \leftarrow Exp_{\mathbf{C}^{[i]}}(\hat{\mathbf{A}}_i)$.
**8.**           $(\hat{\mathbf{C}}^{[i]}, \hat{r}_i) \leftarrow \text{rounding}(\hat{\mathbf{C}}^{[i]})$.
**9.**      $\mathbf{W}^{(t)} \leftarrow \{\hat{\mathbf{C}}^{[1]}, \hat{\mathbf{C}}^{[2]}, \cdots \hat{\mathbf{C}}^{[d]}\}$.
**10.**      $\mathbf{r} \leftarrow (\hat{r}_1, \hat{r}_2, ...\hat{r}_d)$.
**11.**      Reshape the core tensors $\{\mathbf{C}^{[1]}, \mathbf{C}^{[2]}, ..., \mathbf{C}^{[d]}\}$ based on the updated $\mathbf{r}$.
**12.** Return $\mathbf{W}^{(T+1)} = \{\mathbf{C}^{[1]}, \mathbf{C}^{[2]}, \cdots, \mathbf{C}^{[d]}\}$.

---

Similarly, $\mu$ induces the norm of $\mathbf{u} \in T_{\boldsymbol{x}}M$ as $||\mathbf{u}|| = \sqrt{\mu_{\boldsymbol{x}}(\mathbf{u}, \mathbf{u})} \geq 0$. In addition, the $\mu$ induced inner product and the norm preserve the basic properties like definiteness, homogeneity and triangle inequality.

Algorithm 1 presents the RSGD Algorithm. The algorithm mainly consists of two main procedures: one is the update of parameters in the tangent space and conducting an exponential mapping, and the second one is the rounding to lower Tensor-Train ranks. As illustrated in Figure 1, step 5 firstly obtains a gradient $\mathbf{g}_{\mathbf{C}^{[i]}}$ on a tangent space $T_{\mathbf{C}_i} M$ at the core tensor $\mathbf{C}^{[i]}$ in Riemannian Manifold $(M, \mu)$, and step 6 conducts a gradient descent on the tangent space to generate a new tensor $\hat{\mathbf{A}}_i$. Step 7 projects $\hat{\mathbf{A}}_i$ back to $\hat{\mathbf{C}}^{[i]}$ in Riemannian Manifold $(M, \mu)$ by an exponential mapping. Finally, as shown in Figure 2, the rounding function in step 8 transforms $\hat{\mathbf{C}}^{[i]}$ in the submanifold $S_{\mathbf{r}}$ to the core tensor $\hat{\mathbf{C}}^{[i]}$ with a vector of lower Tensor-Train ranks $\hat{\mathbf{r}}$ in a new submanifold $S_{\hat{\mathbf{r}}}$. Note that the vectors of Tensor-Train ranks $\mathbf{r}$ and $\hat{\mathbf{r}}$ span two submanifolds $S_{\mathbf{r}} \subset M$ and $S_{\hat{\mathbf{r}}} \subset M$ respectively. After that, the next iteration of the the parameter update is conducted in $S_{\hat{\mathbf{r}}}$.

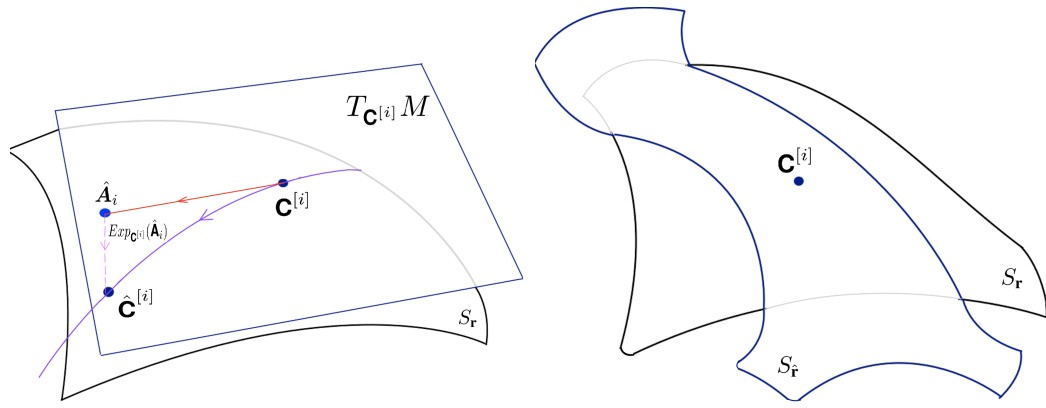

Figure 1: An exponential mapping.          Figure 2: The rounding procedure.

The exponential mapping in Algorithm 1 is formulated in (6). Unfortunately, it is not easy to solve the problem because we have to deal with the calculus of variations, or we have to know the Christoffel symbols Lubich et al. (2015). Therefore, a fast and straightforward retraction method is applied as a first-order approximation to the exponential mapping as shown in Algorithm 2.

$$\min_{\mathbf{Y} \in S_{\mathbf{r}} \subset M^d} ||\mathbf{W} - \mathbf{Y}||_F$$
$$s.t., tt\_rank(\mathbf{Y}) = \mathbf{r}. \tag{6}$$

---

**Algorithm 2** The Retraction Algorithm

---

**1.** Given the core tensors $\{\mathbf{C}^{[1]}, \mathbf{C}^{[2]}, \cdots, \mathbf{C}^{[d]}\}$ and the Tensor-Train rank $\hat{\mathbf{r}} = \{r_1, r_2, \cdots, r_{d+1}\}$.
**2.** For $i = 1$ to $d$:
**3.**     $\boldsymbol{A}_i \leftarrow$ reshape $\mathbf{C}^{[i]}$ by the shape of $(r_i \cdot n_i \cdot m_i) \times r_{i+1}$.
**4.** For $i = 1$ to $d$:
**5.**     $\hat{\boldsymbol{A}}_i, \Sigma_i \leftarrow$ QR_decomposition($\boldsymbol{A}_i$).
**6.**     $\mathbf{C}^{[i]} \leftarrow$ reshape $\hat{\boldsymbol{A}}_i$ by $(r_i, n_i, m_i, r_{i+1})$.
**7.**     $\boldsymbol{A}_{i+1} \leftarrow$ multiply($\Sigma_i, \boldsymbol{A}_{i+1}$),    if $i < d$.
**8.** Return $\{\mathbf{C}^{[1]}, \mathbf{C}^{[2]}, \cdots, \mathbf{C}^{[d]}\}$.

---

**Algorithm 3** The Rounding Algorithm

---

**1.** Given the core tensors $\{\mathbf{C}^{[1]}, \mathbf{C}^{[2]}, \cdots, \mathbf{C}^{[d]}\}$ and a constant maximum rank $r_{\max}$.
**2.** Initialize $\hat{r} = \{1, 1, ..., 1\}$.
**3.** For $i = 1$ to $d$:
**4.**     $\boldsymbol{A}_i \leftarrow$ reshape $\mathbf{C}^{[i]}$ by the shape of $(r_i \cdot n_i \cdot m_i) \times r_{i+1}$.
**5.** For $i = 1$ to $d$:
**6.**     $col\_num(\mathbf{C}^{[i]}) = n_i \cdot m_i \cdot \hat{r}_{i+1}$.
**7.**     $row\_num(\mathbf{C}^{[i]}) = \dfrac{r_i \cdot r_{i+1}}{\hat{r}_{i+1}}$.
**8.**     $\hat{r}_{i+1} = \min(r_{\max}, col\_num(\mathbf{C}^{[i]}), row\_num(\mathbf{C}^{[i]}))$.
**9.**     $(\boldsymbol{U}_i, \boldsymbol{S}_i, \boldsymbol{V}_i) = \text{SVD}(\boldsymbol{A}_i)$.
**10.**     $\hat{\boldsymbol{U}}_i = \boldsymbol{U}_i[:, 1 : \hat{r}_{i+1}], \hat{\boldsymbol{S}}_i = \boldsymbol{S}_i[1 : \hat{r}_{i+1}, 1 : \hat{r}_{i+1}], \hat{\boldsymbol{V}}_i = \boldsymbol{V}_i[1 : \hat{r}_{i+1}, :]$.
**11.**     $\hat{\boldsymbol{A}}_i = \hat{\boldsymbol{U}}_i \hat{\boldsymbol{S}}_i \hat{\boldsymbol{V}}_i$.
**12.**     $\mathbf{C}^{[i]} \leftarrow$ reshape $\hat{\boldsymbol{A}}_i$ as $R^{\hat{r}_i \times n_i \times m_i \times \hat{r}_{i+1}}$.
**13.** Return the updated core tensors $\{\mathbf{C}^{[1]}, \mathbf{C}^{[2]}, \cdots, \mathbf{C}^{[d]}\}$.

---

Algorithm 2 presents the retraction algorithm. The main idea of the retraction algorithm is to orthogonalize the core tensors $\{\mathbf{C}^{[1]}, \mathbf{C}^{[2]}, \cdots, \mathbf{C}^{[d]}\}$ in a left-to-right order by the QR decomposition.

The rounding algorithm is shown in Algorithm 3. Similar to the retraction algorithm, a left-to-right tensor matricization is firstly initialized. Then, the Tensor-Train rank is updated before conducting an SVD computation. The returned core tensors $\{\mathbf{C}^{[1]}, \mathbf{C}^{[2]}, \cdots, \mathbf{C}^{[d]}\}$ are based on the updated vector of lower Tensor-Train ranks. Furthermore, the rounding procedure has the property presented in Proposition 1.

**Proposition 1.** *The rounding procedure of Algorithm 3 does not change values of the objective function $f$ for the Tensor-Train RNNs. That is, for a tensor $\boldsymbol{x} \in S_{\boldsymbol{r}}$, the rounding tensor $\hat{\boldsymbol{x}} \in S_{\hat{\boldsymbol{r}}}$, we have $f(\hat{\boldsymbol{x}}) = f(\boldsymbol{x})$.*

*Proof.* Given the weight matrix $\boldsymbol{W}$ with core tensors $\{\mathbf{C}^{[1]}, \mathbf{C}^{[2]}, \cdots, \mathbf{C}^{[d]}\}$, for an input tensor $\mathbf{X} \in M$, we obtain (7) and (8) according to (1).

$$\mathbf{Y}(j_1, j_2, ..., j_d) = \sum_{i_1=1}^{m_1} \cdots \sum_{i_d=1}^{m_d} \prod_{t=1}^{d} \mathbf{C}^{[t]}(r_t, i_t, j_t, r_{t+1}) \mathbf{X}(i_1, i_2, ..., i_d) + \mathbf{B}(j_1, j_2, ..., j_d) \quad (7)$$

$$= \sum_{i_1=1}^{m_1} \cdots \sum_{i_d=1}^{m_d} \prod_{t=1}^{d} \mathbf{C}^{[t]}(\hat{r}_t, i_t, j_t, \hat{r}_{t+1}) \mathbf{X}(i_1, i_2, ..., i_d) + \mathbf{B}(j_1, j_2, ..., j_d), \quad (8)$$

which suggests that a vector of Tensor-Train ranks determines a submanifold for generating core tensors of the tensor $\mathbf{W}$, but the values of the objective functions are invariant to the change of the vector of Tensor-Train ranks, obtaining Proposition 1.                                   □

## 3 CONVERGENCE ANALYSIS

This section analyzes the convergence of the RSGD algorithm. The necessary definitions and theorems are firstly introduced, and the analysis is then provided. Since the objective functions of Tensor-Train RNNs are always geodesically non-convex Zhang et al. (2016), we only consider the convergence of RSGD for non-convex cases.

**Definition 2.** *A different function $f : M \to R$ is geodesically L-smooth if its first-order gradient is geodesically L-Lipchitz continuous. Specifically, $\forall x, y \in M$ we have 9.*

$$f(\boldsymbol{y}) \leq f(\boldsymbol{x}) + <g_{\boldsymbol{x}}, Exp_{\boldsymbol{x}}^{-1}(\boldsymbol{y}) > + \frac{L}{2}||Exp_{\boldsymbol{x}}^{-1}(\boldsymbol{y})||^2, \tag{9}$$

where $g_{\boldsymbol{x}}$ is the sub-gradient of $f(\boldsymbol{x})$ at $\boldsymbol{x}$ in the tangent space $T_{\boldsymbol{x}}M$, and $Exp_{\boldsymbol{x}}^{-1}(\boldsymbol{y})$ is the inverse exponential mapping which projects the curve line in $M^d$ from $\boldsymbol{x}$ to $\boldsymbol{y}$ back to the gradient $\boldsymbol{x}$ in the tangent space $T_{\boldsymbol{x}}M$.

From the RSGD algorithm (Algorithm 1), it is not hard to find the sub-gradient $g_{\boldsymbol{x}} = \nabla f(\boldsymbol{x})$ and $Exp_{\boldsymbol{x}}^{-1}(\boldsymbol{y}) = -\eta \nabla_{\boldsymbol{x}} f(\boldsymbol{x})$, and thus Theorem 3 can be derived.

**Theorem 3.** *For a differentiable and geodesically L-smooth function $f$, the Riemannian stochastic gradient descent algorithm ensures (10).*

$$\min_t E[||\nabla f(\boldsymbol{x}_t)|||] \leq \frac{1}{\eta T}(f(\boldsymbol{x}_0) - f(\boldsymbol{x}^*)) + \frac{L}{2}\eta^2 k^2, \tag{10}$$

where $T$ refers the total iterations, $\boldsymbol{x}_0$ and $\boldsymbol{x}^*$ denote the initial and the optimal points respectively, $\eta$ is the learning rate, and $||Exp_{\boldsymbol{x}}^{-1}(\boldsymbol{y})||^2$ is bounded by $\eta^2 k^2$.

*Proof.* Assume $x^*$ and $x_0$ separately refer to the optimal and initial points. For all $x_t$ and $x_{t+1}$ at two consecutive times, we can derive (11) based on Definition 2.

$$E[f(\boldsymbol{x}_{t+1})] \leq E[f(\boldsymbol{x}_t)] + E[< \nabla f(\boldsymbol{x}_t), Exp_{\boldsymbol{x}_t}^{-1}(\boldsymbol{x}_{t+1}) >] + \frac{L}{2}E[||Exp_{\boldsymbol{x}_t}^{-1}(\boldsymbol{x}_{t+1})||^2]. \tag{11}$$

By applying $Exp_{\boldsymbol{x}_t}^{-1}(\boldsymbol{x}_{t+1}) = -\eta \nabla f(\boldsymbol{x}_t))$ and $E[||Exp_{\boldsymbol{x}_t}^{-1}(\boldsymbol{x}_{t+1})||^2] \leq \eta^2 k^2$, we obtain (12), where $\hat{\boldsymbol{x}}_{t+1}$ is the rounding tensor of $\hat{\boldsymbol{x}}_{t+1}$. The Proposition 1 ensures that $f(\boldsymbol{x}_{t+1}) = f(\hat{\boldsymbol{x}}_{t+1})$.

$$E[||\nabla f(\boldsymbol{x}_t)|||] \leq \frac{1}{\eta}E[f(\boldsymbol{x}_t) - f(\boldsymbol{x}_{t+1})] + \frac{L}{2}\eta^2 k^2 = \frac{1}{\eta}E[f(\boldsymbol{x}_t) - f(\hat{\boldsymbol{x}}_{t+1})] + \frac{L}{2}\eta^2 k^2. \tag{12}$$

By summing the two sides of equation 12 from 0 to $T - 1$, we derive (13).

$$\min_t E[||\nabla f(\boldsymbol{x}_t)|||] \leq \frac{1}{T}\sum_{t=0}^{T-1} E[||\nabla f(\boldsymbol{x}_t)|||] \leq \frac{1}{\eta T}(f(\boldsymbol{x}_0) - f(\boldsymbol{x}_T)) + \frac{L}{2}\eta^2 k^2. \tag{13}$$

Since $f(\boldsymbol{x}_t) \leq f(\boldsymbol{x}_T), \forall 0 \leq t \leq T - 1$, we have 14.

$$\min_t E[||\nabla f(\boldsymbol{x}_t)|||] \leq \frac{1}{\eta T}(f(\boldsymbol{x}_0) - f(\boldsymbol{x}_T)) + \frac{L}{2}\eta^2 k^2 \leq \frac{1}{\eta T}(f(\boldsymbol{x}_0) - f(\boldsymbol{x}^*)) + \frac{L}{2}\eta^2 k^2. \tag{14}$$

After rounding $\boldsymbol{x}^*$ to $\hat{\boldsymbol{x}}^*$ , we finally obtain the result 15 of Theorem 3.

$$\min_t E[||\nabla f(\boldsymbol{x}_t)|||] \leq \frac{1}{\eta T}(f(\boldsymbol{x}_0) - f(\hat{\boldsymbol{x}}^*)) + \frac{L}{2}\eta^2 k^2. \tag{15}$$

$\square$

Furthermore, Theorem 3 suggests that the number of iterations $T$ satisfies (16) before reaching the convergence.

$$T \leq \frac{f(\boldsymbol{x}_0) - f(\hat{\boldsymbol{x}}^*)}{\eta(\min_t E[||\nabla f(\boldsymbol{x}_t)|||] - \frac{L}{2}\eta^2 k^2)}. \tag{16}$$

# 4 ADVANCED TENSOR-TRAIN RNNS

This section introduces the Tensor-Train RNNs with the advanced architectures used in this work. The new Tensor-Train architecture is based on the Bi-directional Tensor-Train GSU/LSTM. The other one is the Tensor-Train Encoder-Decoder RNNs with the Tensor-Train Attention mechanism.

## 4.1 BI-DIRECTIONAL TENSOR-TRAIN GRU/LSTM

Firstly, we introduce the Bi-directional Tensor-Train GRU/LSTM, which involves twice more parameters than the Tensor-Train GRU/LSTM. Equations from (17) to (24) present the functional mechanisms of the Bi-directional Tensor-Train GRU, where the pairs $(\overrightarrow{r_t}, \overleftarrow{r_t})$ and $(\overrightarrow{z_t}, \overleftarrow{z_t})$ denote forward and backward update gate operations, respectively, and the pair $(\overrightarrow{d_t}, \overleftarrow{d_t})$ represents the forward-backward reset gate operations. In addition, the pair $(\overrightarrow{h_t}, \overleftarrow{h_t})$ is the memory cell holding information of last times. $o_h$ is the output of the Bi-directional Tensor-Train GRU which combines $\overrightarrow{h_t}$ and $\overleftarrow{h_t}$ by concatenating them before feeding through a Tensor-Train linear layer.

$$\overrightarrow{r_t} = \sigma(TTL(\overrightarrow{\boldsymbol{W}}^r, \overrightarrow{\boldsymbol{X}_t}) + TTL(\overrightarrow{\boldsymbol{U}}^r, \overrightarrow{\boldsymbol{h}_{t-1}})) \tag{17}$$

$$\overleftarrow{r_t} = \sigma(TTL(\overleftarrow{\boldsymbol{W}}^r, \overleftarrow{\boldsymbol{X}_t}) + TTL(\overleftarrow{\boldsymbol{U}}^r, \overleftarrow{\boldsymbol{h}_{t-1}})) \tag{18}$$

$$\overrightarrow{z_t} = \sigma(TTL(\overrightarrow{\boldsymbol{W}}^z, \overrightarrow{\boldsymbol{X}_t}) + TTL(\overrightarrow{\boldsymbol{U}}^z, \overrightarrow{\boldsymbol{h}_{t-1}})) \tag{19}$$

$$\overleftarrow{z_t} = \sigma(TTL(\overleftarrow{\boldsymbol{W}}^z, \overleftarrow{\boldsymbol{X}_t}) + TTL(\overleftarrow{\boldsymbol{U}}^z, \overleftarrow{\boldsymbol{h}_{t-1}})) \tag{20}$$

$$\overrightarrow{d_t} = tanh(TTL(\overrightarrow{\boldsymbol{W}}^d, \overrightarrow{\boldsymbol{X}_t}) + TTL(\overrightarrow{\boldsymbol{U}}^d, (\overrightarrow{r_t} \circ \overrightarrow{\boldsymbol{h}_{t-1}}))) \tag{21}$$

$$\overleftarrow{d_t} = tanh(TTL(\overleftarrow{\boldsymbol{W}}^d, \overleftarrow{\boldsymbol{X}_t}) + TTL(\overleftarrow{\boldsymbol{U}}^d, (\overleftarrow{r_t} \circ \overleftarrow{\boldsymbol{h}_{t-1}}))) \tag{22}$$

$$\overrightarrow{h_t} = (1 - \overrightarrow{z_t}) \circ \overrightarrow{\boldsymbol{h}_{t-1}} + \overrightarrow{z_t} \circ \overrightarrow{d_t}, \quad \overleftarrow{h_t} = (1 - \overleftarrow{z_t}) \circ \overleftarrow{\boldsymbol{h}_{t-1}} + \overleftarrow{z_t} \circ \overleftarrow{d_t} \tag{23}$$

$$o_h = \sigma(TTL(\boldsymbol{A}_h, [\overrightarrow{h_t}; \overleftarrow{h_t}])). \tag{24}$$

In addition, we also design a Bi-directional Tensor-Train LSTM which involves more operational gates than the Bi-directional Tensor-Train GRU.

## 4.2 THE TENSOR-TRAIN ENCODER-DECODER RNNS WITH ATTENTION MODELS

The Encoder-Decoder RNNs are commonly used in sequence-to-sequence deep learning applications. Moreover, the attention mechanism significantly improves the performance of Encoder-Decoder RNNs Vaswani et al. (2017).

The Bi-directional Tensor-Train RNN like GRU or LSTM is used to construct the Encoder-Decoder architecture. Moreover, we set up the Tensor-Train Attention model in addition to the Tensor-Train Encoder-Decoder RNNs. Thus, the entire model is built on Tensor-Train layers.

To build a Tensor-Train Attention model, it is necessary to add the Tensor-Train layer to generate an Attention vector as shown in (25), where $\mathbf{c}_t$ is a context vector (26) with attention weights $\alpha_{ts}$ (27), $\boldsymbol{a}_t$ is the output of the Attention model at time $t$, $\bar{\boldsymbol{h}}_s$ denotes a vector built from the outputs of the forward and backward stages, and $\boldsymbol{h}_t$ refers to the output of the hidden layers at time $t$.

$$\boldsymbol{a}_t = tanh(TTL(W_c, [\boldsymbol{c}_t; \boldsymbol{h}_t])) \tag{25}$$

$$\boldsymbol{c}_t = \sum_s \alpha_{ts} \bar{\boldsymbol{h}}_s \tag{26}$$

$$\alpha_{ts} = \frac{\exp(score(\boldsymbol{h}_t, \bar{\boldsymbol{h}}_s))}{\sum_{s'=1}^{S} \exp(score(\boldsymbol{h}_t, \bar{\boldsymbol{h}}_{s'}))}. \tag{27}$$

# 5 APPLICATIONS

This section first introduces the implementation of the Tensor-Tensor RNNs. Then, we present two applications where the RSGD algorithms were tested. One application is the digit recognition task on the sequential MNIST dataset; the other is the task of machine translation on the Multi30K dataset Elliott et al. (2016).

## 5.1 IMPLEMENTATIONS

We employed PyTorch to implement our Tensor-Train RNNs. The data structures of our implementations were partly built on the free Tensor-Train toolkit implemented by the tool Tensorflow Novikov et al. (2018). However, we employed the tool PyTorch to take the advantage of dynamic graph generation, which is much more useful for NLP tasks.

## 5.2 DIGIT RECOGNITION ON THE MNIST DATASET

The first application is the digit recognition task on the MNIST dataset. The dataset consists of $60000$ data with $28 * 28$ pixels for each digital image. Instead of vectorizing the image pixels into a long vector as an input for a static deep neural network, image pixels are taken as data sequence where the time step is set to $28$, and the input dimension is set to $28$. In our experiments, the training and testing sets were separately composed of $50000$ and $10000$ data. $2000$ data were selected from the training set for building a validation set, and they were not included in the training set.

As for the experimental setup, we applied both Bi-directional Tensor-Train GRU and Bi-directional Tensor-Train LSTM to the task. The dimension of the Manifold $(M, \mu)$ was set to $d = 3$, and the vector of Tensor-Train ranks was initialized with high and shared values $\mathbf{r} = (1, 10, 10, 1)$ for the Tensor-Train layers. The weight matrix of the input-hidden Tensor-Train layer was converted to the tensor with the shape $(2 \times 7 \times 2)$ by $(6 \times 6 \times 6)$, and the weight matrix of the hidden-hidden Tensor-Train layer was converted to the tensor with the shape $(6 \times 6 \times 6)$ by $(6 \times 6 \times 6)$. The RSGD algorithm with a learning rate $0.01$ was applied to both Bi-directional Tensor-Train GRU/LSTM.

The results are shown in Figure 3, where we compared the Bi-directional Tensor-Train GRU/LSTM with the traditional Bi-directional GRU/LSTM regarding recognition error rates and number of parameters. Inspecting the recognition error rate, the Bi-directional Tensor-Train GRU obtains a result that is close to that of the traditional Bi-directional GRU/LSTM, and the performance of the Bi-directional Tensor-Train LSTM becomes a bit worse. Regarding comparison with the number of parameters, both Bi-directional Tensor-Train GRU/LSTM can significantly reduce the number of parameters by taking only $1\%$ parameters of the Bi-directional GRU/LSTM at final. Notably, the RSGD algorithm further reduces the number of parameters of Bi-directional GRU/LSTM by lowering the Tensor-Train ranks.

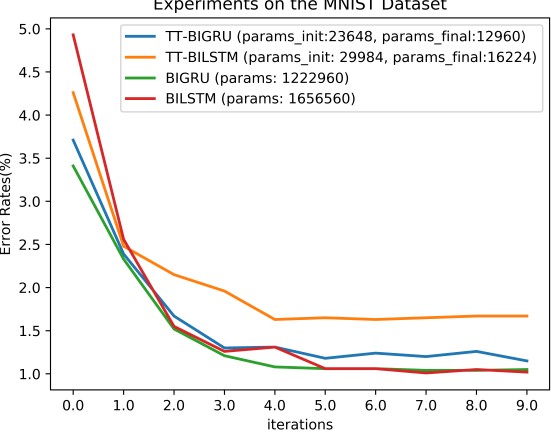

Figure 3: Experimental results on the sequential MNIST dataset ('TT' denotes Tensor-Train, 'params.' means the number of parameters, 'init.' means the initial, and 'final' refers to the final number).

Table 1: Statistics of Model Parameters

| Models | Parameters of hidden layers |
|---|---|
| Bi-directional Encoder-Decoder | 462144 |
| Tensor-Train Bi-directional Encoder-Decoder (initial) | 15154 |
| Tensor-Train Bi-directional Encoder-Decoder (final) | 11088 |

### 5.3 MACHINE TRANSLATION ON THE MULTI30K DATASET

The next application is a machine translation task from Dutch to English on the Multi30K dataset. In the dataset, there are separately 29000, 1014, and 1000 sentence pairs for training data, validation data, and test data, respectively.

The Bi-directional Tensor-Train GRU was used to build the Encoder-Decoder architecture, and a Tensor-Train Attention model was added to the architecture. The RSGD algorithm with a learning rate $0.01$ was applied to the Tensor-Train Encoder-Decoder RNNs with the Tensor-Train Attention model. An initial vector of Tensor-Train ranks was set as $\mathbf{r} = (1, 6, 6, 6, 1)$, and the weight matrix of hidden layers was converted to the tensor with the shape of $(4 \times 4 \times 4 \times 4)$ by $(4 \times 4 \times 4 \times 4)$. Besides, the baseline was based on the traditional Bi-directional GRU and LSTM-based Encoder-Decoder architecture with the Attention model, where the Stochastic Gradient Descent algorithm with learning rate $0.001$ was used to update parameters.

The experimental results are shown in Figure 4, and the statistics of parameters of hidden layers are shown in Table 1. The results in Figure 4 suggest that the Tensor-Train Encoder-Decoder RNN performs closer or even better than the Encoder-Decoder one, although the convergence speed of the Tensor-Train model is relatively slower in the first several iterations. On the other hand, Table 1 shows that RSGD leads to a further decrease in the number of parameters.

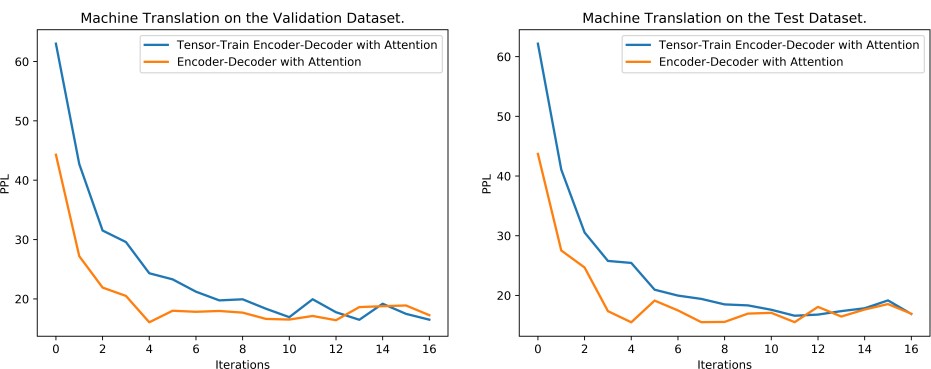

Figure 4: Experimental results of Machine Translation on the Multi30K dataset (PPL refers to the logarithm of loss values).

## 6 CONCLUSIONS

This paper presents the RSGD algorithm for training Tensor-Train RNNs including the related properties, implementations, and convergence analysis. Our experiments on digit recognition and machine translation tasks suggest that RSGD can work effectively on the Tensor-Train RNNs regarding performance and model complexity, although the convergence speed is relatively slower in the beginning stages. Our future work will consider two directions: one is to apply the RSGD algorithm to more Tensor-Train models and test it on larger datasets of other fields; and the second one is to generalize Riemannian optimization to the variants of the SGD algorithms and study how to speed up the convergence rate.

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
