# OpenReview forum: "Riemannian Stochastic Gradient Descent for Tensor-Train Recurrent Neural Networks"
_ICLR.cc/2019/Conference_

### Official Review · AnonReviewer1 · 2018-10-30
**A paper on Riemannian optimization, needs to fix some math and improve experiments**

**Rating:** 3
**Confidence:** 3

**Review:**

This paper proposes an algorithm for optimizing neural networks parametrized by Tensor Train (TT) decomposition based on the Riemannian optimization and rank adaptation, and designs a bidirectional TT LSTM architecture.

I like the topic chosen by the authors, using TT to parametrize layers of neural networks proved to be beneficial and it would be very nice to exploit the Riemannian manifold structure to speed up the optimization.

But, the paper needs to be improved in several aspects before being useful to the community. In particular, I found the several mathematical errors regarding basic definitions and algorithms (see below the list of problems) and I’m not happy with lack of baselines in the experimental comparison (again, see below).

The math problems
1) In equations (1), (2), (7), and (8) there is an error: one should sum out the rank dimensions instead of fixing them to the numbers r_i. At the moment, the left-hand side of the equations doesn’t depend on r and the right-hand side does.
2) In two places the manifold of d-dimensional low-rank tensors is called d-dimensional manifold which is not correct. The tensors are d-dimensional, but the dimensionality of the manifold is on the order of magnitude of the number of elements in the cores (slightly smaller actually).
3) The set of tensors with rank less than or equal to a fixed rank (or a vector of ranks) doesn’t form a Riemannian (or smooth for that matter) manifold. The set of tensors of rank equal to a fixed rank something does.
4) The function f() minimized in (5) is not defined (it should be!), but if it doesn’t have any rank regularizer, then there is no reason for the solution of (5) to have rank smaller then r (and thus I don’t get how the automatic rank reduction can be done).
5) When presenting a new retraction algorithm, it would be nice to prove that it is indeed a retraction. In this case, Algorithm 2 is almost certainly not a retraction, I don’t even see how can it reduce the ranks (it has step 6 that is supposed to do it, but what does it mean to reshape a tensor from one shape to a shape with fewer elements?).
6) I don’t get step 11 of Alg 1, but it seems that it also requires reshaping a tensor (core) to a shape with fewer elements.
7) The rounding algorithm (Alg 3) is not correct, it has to include orthogonalization (see Oseledets 2011, Alg 2).
8) Also, I don’t get what is r_max in the final optimization algorithm (is it set by hand?) and how the presented rounding algorithm can reduce the rank to be lower than r_max (because if it cannot, one would get the usual behavior of setting a single value of rank_max and no rank adaptivity).
9) Finally, I don’t get the proposition 1 nor it’s proof: how can it be that rounding to a fixed r_max won’t change the value of the objective function? What if I set r_max = 1? We should be explained in much greater detail.
10) I didn’t get this line: “From the RSGD algorithm (Algorithm 1), it is not hard to find the sub-gradient gx = ∇f(x) and Exp−1 x (y) = −η∇xf(x), and thus Theorem 3 can be derived.” What do you mean that it is not hard to find the subgradient (and what does it equal to?) and why is the inverse of the exponential map is negative gradient?
11) In general, it would be beneficial to explain how do you compute the projected gradient, especially in the advanced case. And what is the complexity of this projection?
12) How do you combine optimizing over several TT objects (like in the advanced RNN case) and plain tensors (biases)? Do you apply Riemannian updates independently to every TT objects and SGD updates to the non-TT objects? Something else?
13) What is E in Theorem 3? Expected value w.r.t. something? Since I don’t understand the statement, I was not able to check the proof.

The experimental problems:
1) There is no baselines, only the vanilla RNN optimized with SGD and TT RNN optimized with your methods. There should be optimization baseline, i.e. optimizing the same TT model with other techniques like Adam, and compression baselines, showing that the proposed bidirectional TT LSTM is better than some other compact architectures. Also, the non-tensor model should be optimized with something better than plain SGD (e.g. Adam).
2) The convergence plots are shown only in iteration (not in wall clock time) and it’s not-obvious how much overhead the Riemannian machinery impose.
3) In general, one can decompose your contributions into two things: an optimization algorithm and the bidirectional TT LSTM. The optimization algorithm in turn consist in two parts: Riemannian optimization and rank adaptation. There should be ablation studies showing how much of the benefits come from using Riemannian optimization, and how much from using the rank adaptation after each iteration.

And finally some typos / minor concerns:
1) The sentence describing the other tensor decomposition is a bit misleading, for example CANDECOMP can also be scaled to arbitrary high dimensions (but as a downside, it doesn’t allow for Riemannian optimization and can be harder to work with numerically).
2) It’s very hard to read the Riemannian section of the paper without good knowledge of the subject, for example concepts of tangent space, retraction, and exponential mapping are not introduced.
3) In Def 2 “different function” should probably be “differentiable function”.
4) How is W_c represented in eq (25), as TT or not? It doesn’t follow the notation of the rest of the paper. How is a_t used?
5) What is “score” in eq (27)?
6) Do you include bias parameters into the total number of parameters in figures?
7) The notation for tensors and matrices are confusingly similar (bold capital letters of slightly different font).
8) There is no Related Work section, and it would be nice to discuss the differences between this work and some relevant ones, e.g. how is the proposed advanced TT RNN different from the TT LSTMs proposed in Yang et al. 2017 (is it only the bidirectional part that is different?) and how is the Riemannian optimization part different from Novikov et al. 2017 (Exponential machines), and what are the pros and cons of your optimization method compared to the method proposed in Imaizumi et al. 2017 (On Tensor Train Rank Minimization: Statistical Efficiency and Scalable Algorithm).


Please, do take this as a constructive criticism, I would be happy to see you resubmitting the paper after fixing the raised concerns!

---

### Official Review · AnonReviewer3 · 2018-11-04
**The paper presents RSGD algorithm on TT based RNN, which is interesting but the quality and significance is limited.**

**Rating:** 4
**Confidence:** 4

**Review:**

In this paper, the authors proposed a new method to update the weights in RNN by SGD on Riemannian manifold.  Due to the properties of manifold learning, the updated weights in each iteration are contracted with a low-rank structure, such that the number of the parameters of TT can be automatically decreased during the training procedure. By using the new algorithm, the authors modified two types of sophisticated RNNs, i.e., bi-directional GRU/LSTM and Encoder-Decoder RNN. The experimental results validate effectiveness of the proposed method. How to determine the rank of the tensor networks in weight compression problem is indeed an important and urgent task, this paper does not clearly illustrate how RSGD can efficiently solve this problem.

1. Compared to the conventional SGD, not only the convergence rate of the proposed method seems slower (mentioned in the paper,), but also additional computational operations should be done in each iteration like exponential mapping (with multiple QR and SVD). I’m worried about the computational efficiency of this method, but  this paper neither discusss the  computational complexity nor illustrate the results in the experimental section.

2. In proof of proposition 1, I’m confused why the input tensor X should belong to M, and why the eq. (8) holds?

3. In the convergence analysis, I don’t know why the eq.  $Exp^{-1}(y)=-\eta….$ holds even though the authors claims the it is not hard to find. So that, I cannot find the relationship between Theorem 3 and the proposed method.  Furthermore, can Theorem 3 be used to prove the convergence of the proposed method?

4. Eq. (16) would make no sense because the denominator might be very small.

5. In the experiment, please compare with other existing (tensor decomposition based) compression methods to demonstrate how the proposed method makes sense in this task.

Minior:
1. By the definition in Oseledets’ paper, the tensor decomposition model used in this paper should be called TT-matrix rather than TT.
2. 9 ->(9) in Definition 2, and 15->(15) in the proof of Theorem 3.

---

### Official Review · AnonReviewer2 · 2018-11-05
**Novelty is limited**

**Rating:** 4
**Confidence:** 4

**Review:**


Summary:
The paper proposes to use Riemannian stochastic gradient algorithm for low-rank tensor train learning in deep networks.

Comments:
The paper is easy to follow.

C1.
The novelty of the paper is rather limited, both in terms of the convergence analysis and exploiting the low-rank structure in tensor trains. It misses the important reference [1], where low-rank tensor trains have been already discussed. Section 3 is also not novel to the paper. Consequently, Sections 2 and 3 have to be toned down.

Section 4 is interesting but is not properly written. There is no discussion on how the paper comes about those modifications. It seems that the paper blindly tries to apply the low-rank constraint to the works of Chung et al. (2014) and Luong et al. (2015).

[1] https://epubs.siam.org/doi/abs/10.1137/15M1010506
Steinlechner, Michael. "Riemannian optimization for high-dimensional tensor completion." SIAM Journal on Scientific Computing 38.5 (2016): S461-S484.

C2.
The constraint tt_rank(W) \leq r in (5) is not a manifold. The equality is needed for the constraint to be a manifold.

C3.
Use \langle and \rangle for inner products.

---

### Meta-Review · Area_Chair1 · 2018-12-11
**ICLR 2019 decision**

**Confidence:** 5
**Recommendation:** Reject

**Metareview:**

This paper proposes using a tensor train low rank decomposition for compressing neural network parameters.  However the paper falls short on multiple fronts 1)lack of comparison with existing methods 2) no baseline experiments. Further there are concerns about correctness of the math in deriving the algorithms, convergence and computational complexity of the proposed method.  I strongly suggest taking the reviews into account before submitting the paper it again.